# Rates of cardiovascular events among patients with moderate-to-severe atopic dermatitis in an integrated health care system: A retrospective cohort study

Monique M. Hedderson[1], Maryam M. Asgari[2], Fei Xu[1], Charles P. Quesenberry[1], Sneha Sridhar[1], Jamie Geier[3], Adina R. Lemeshow[3]*

1 Kaiser Permanente Northern California, Oakland, California, United States of America, 2 Massachusetts General Hospital, Harvard Medical School, Boston, Massachusetts, United States of America, 3 Pfizer, Inc., New York, New York, United States of America

* Adina.lemeshow@pfizer.com

## Abstract

Patients with versus without atopic dermatitis may have a greater risk of cardiovascular events, and the risk increases with severity of atopic dermatitis. The incidence of cardiovascular events in the population of patients with moderate-to-severe atopic dermatitis is largely unknown. This retrospective study evaluates incidence rates of cardiovascular events in patients aged ≥12 years with moderate-to-severe atopic dermatitis in a cohort of Kaiser Permanente Northern California health care system members without recognized risk factors for adverse events. Patients with moderate-to-severe atopic dermatitis, as defined by dermatologist-rendered code and prescription history between 2007 and 2018, were included. Major adverse cardiovascular events, venous thrombotic events, deep vein thrombosis, and pulmonary embolisms were identified via International Classification of Diseases codes. Stratification variables included age, sex, race, smoking history, and diabetes. Incidence rates per 1000 person-years were calculated by the number of patients with an incident event divided by the total person-years of observation. Among 8197 patients with moderate-to-severe atopic dermatitis, incidence rates per 1000 person-years (95% confidence interval) for major adverse cardiovascular events, venous thrombotic events, deep vein thrombosis, and pulmonary embolism were: 2.6 (2.1–3.2), 2.0 (1.5–2.5), 1.6 (1.2–2.1), and 0.7 (0.5–1.0), respectively. Incidence rates for all events were higher for older versus younger patients, patients with versus without diabetes, former smokers versus patients who had never smoked, and men versus women, except for pulmonary embolisms, which were higher in women. This study estimated the incidence of cardiovascular events in patients with moderate-to-severe atopic dermatitis and provides valuable information for clinicians.

**Data Availability Statement:** Individual level data may not be made publicly available due to IRB and privacy concerns. The data used for this study

contain protected health information (PHI) and access is protected by the Kaiser Permanente Northern California Institutional Review Board (IRB). Data are available from the Kaiser Permanente Division of Research for researchers who meet the criteria for access to confidential data. For more information about data access and criteria for access to confidential data, please contact Kaiser Permanente Division of Research: DOR.IRB.Submissions@kp.org. All other relevant data are within the paper.

**Funding:** This non-interventional study was sponsored by Pfizer, Inc. (https://www.pfizer.com/). Pfizer offered input for consideration in the study design and decision to publish. Pfizer played no role in data collection. Under the guidance and direction of authors, editorial/medical writing support was provided by Marianna Johnson, PhD, and Natalie Barnes, BASLP, at ApotheCom, San Francisco, CA and was funded by Pfizer. Monique M. Hedderson, Fei Xu, Charles P. Quesenberry, and Sneha Sridhar are full-time employees of Kaiser Permanente Northern California and received grant funding from Pfizer Inc. for the study discussed in this publication. Jamie Geier is a former employee of and shareholder of Pfizer Inc. Adina R. Lemeshow is an employee and shareholder of Pfizer Inc. Maryam M. Asgari received royalty payments from UptoDate for publications related to skin cancer and received grant funding to her institution from Pfizer Inc. for the study described in this publication (no grant number is applicable).

**Competing interests:** Monique M. Hedderson, Fei Xu, Charles P. Quesenberry, and Sneha Sridhar are full-time employees of Kaiser Permanente Northern California and received grant funding from Pfizer Inc. for the study discussed in this publication. Jamie Geier is a former employee of and shareholder of Pfizer Inc. Adina R. Lemeshow is an employee and shareholder of Pfizer Inc. Maryam M. Asgari received royalty payments from UptoDate for publications related to skin cancer and received grant funding to her institution from Pfizer Inc. for the study described in this publication.

## Introduction

Atopic dermatitis (AD) is a chronic inflammatory skin disease characterized by intense itch, red scaly patches, and dry skin [1]. Typically recognized as a childhood disease, there is growing evidence of a high prevalence (up to 23%) and disease burden of AD in adults worldwide [2,3]. Patients with AD often experience diminished health-related quality of life owing to debilitating chronic itch, sleep disturbance, and psychological distress, with increased AD severity associated with worsening health-related quality of life [4–7]. Despite available AD management strategies, including topical or systemic therapies, a significant proportion of patients with AD continues to have inadequately controlled disease and persistent risk for multiple comorbidities, including atopic (e.g., asthma, allergic rhinitis, and food allergies) and non-atopic (e.g., obesity, high blood pressure, diabetes, and depression/anxiety) conditions [4,8].

Cardiovascular (CV) events have also been associated with uncontrolled AD, independent of genetic risk, and may be due to the complex relationship with immune-mediated inflammation in AD [8–16]. Although the risk of CV outcomes has been evaluated in the AD patient population [17], data on incidence rates (IRs) of CV events among patients with moderate-to-severe AD are limited. Increasing AD severity has been associated with an increase in risk for adverse CV outcomes, such as angina, myocardial infarction, coronary revascularization, heart failure, cardiac arrhythmias, and stroke, suggesting that patients with moderate-to-severe AD may be at greater CV risk than the broader AD population [9]. Given the growing epidemiologic burden of AD worldwide, there is a critical need to assess the incidence of CV in the AD patient population to inform clinical decision making and contextualize events in ongoing clinical trials.

This retrospective cohort study evaluated the IRs of CV events in patients (aged ≥12 years) with moderate-to-severe AD in a cohort of Kaiser Permanente Northern California (KPNC) health plan members.

## Materials and methods

This study was conducted in accordance with legal and regulatory requirements, as well as with scientific purpose, value, and rigor, and followed generally accepted research practices described in Good Pharmacoepidemiology Practices, Best Practices for Conducting and Reporting Pharmacoepidemiologic Safety Studies Using Electronic Healthcare Data, Council for International Organizations of Medical Sciences International Ethical Guidelines for Epidemiological Studies, and the European Network of Centres for Pharmacoepidemiology and Pharmacovigilance Guide on Methodological Standards in Pharmacoepidemiology. This study was approved by the KPNC Institutional Review Board (IRB). A waiver of written informed consent to use patients' electronic medical record data was provided by the IRB; thus, patients did not provide informed consent for use of their data in this study. While Pfizer is a commercial funder of this study, at no point did this interfere with the full and objective presentation, peer review, editorial decision-making, or publication of research data submitted to PLOS in this manuscript. Pfizer products are not evaluated in this study, and no medical value claims are made regarding Pfizer products. Pfizer and all authors declare the results of this study will not be used for any medication that Pfizer is working to submit to the regulatory authorities as part of a new drug application, including the US Food and Drug Administration.

### Study design

The study was conducted by identifying and characterizing a cohort of KPNC health plan members with moderate-to-severe AD from the KPNC database. KPNC is an integrated health care delivery system that provides comprehensive health care and pharmaceutical benefits to

approximately 30% of a large and diverse community-based population residing in Northern California. The sociodemographic and health characteristics in the service area are very similar to the general population in Northern California [18]. The KPNC computerized record system contains administrative and clinical electronic databases, including information on each outpatient prescription dispensed at a KPNC pharmacy.

## Eligibility criteria

The cohort of KPNC health plan members included in this study was aged ≥12 years and were required to have a clinical diagnosis of moderate-to-severe AD (defined as having an outpatient dermatologist-rendered visit with an International Classification of Diseases, Ninth Revision [ICD-9] code of 691.8 or an ICD-10 code of L20 between January 1, 2007, and December 31, 2018). Patients were only included if they enrolled in a KPNC health plan ≥1 year before the study index date. The index date was defined as the first date an eligible health plan member filled a prescription for a topical or systemic agent or received phototherapy. Moderate AD cases were defined as necessitating prescribed topical therapy or phototherapy, and severe AD cases were defined as necessitating prescribed systemic treatment (i.e., oral corticosteroids, cyclosporine, azathioprine, methotrexate, mycophenolic acid/mycophenolic mofetil, interferon-gamma, alitretinoin, rituximab, ustekinumab, omalizumab, dupilumab, or apremilast).

Health plan members were excluded from the study if they had any prior documented evidence of any malignancy (except non-melanoma skin cancer or cervical carcinoma *in situ*), hepatitis B or C infection, or HIV infection; tuberculosis or active inflammatory skin disease and use of any Janus kinase inhibitors within 365 days before the index date; or systemic infections that required hospitalization or parenteral anti-microbial therapy within 6 months of the index date. These criteria were applied to reflect recent clinical trial populations in studies of AD treatments and to reduce the likelihood that CV events observed after the index date were caused by underlying conditions or concomitant drug use. The complete list of exclusion criteria is shown in S1 Table.

## Case adjudication

A medical record review of 100 cases (50 moderate and 50 severe AD cases) was conducted by a board-certified dermatologist (author MMA) to adjudicate and confirm the classification of AD, including AD severity based on established diagnostic criteria [19]. Information on personal and family history of atopy, including asthma and allergic rhinitis as documented in the electronic medical record, was also examined and assigned case status. A "definite" case of AD was defined as a health plan member with documented diagnostic features in the description of clinical examination findings. A "possible" case was defined as a member with some skin findings suggestive of AD but insufficient to assign the case as definite. "Not a case" was defined as a member without evidence for skin or symptom descriptions that indicated AD or, if upon review of the electronic medical record, subsequent biopsy-proven confirmation of a different dermatologic diagnosis (an illustrative example is severe AD that was subsequently diagnosed as cutaneous T cell lymphoma). Cases were excluded if identified as "not a moderate-to-severe case," including mild AD cases, from the analytic cohort. *A priori*, a threshold of ≥80% agreement between the cases identified by study parameters and medical record review was deemed acceptable for disease severity classification.

## Outcomes

CV events (major adverse cardiovascular events [MACEs], venous thrombotic events [VTEs], deep vein thrombosis [DVT], and pulmonary embolisms [PEs]) were analyzed regardless of

presumed causality. They were graded in terms of severity and coded using the Medical Dictionary for Regulatory Activities (S2 Table). MACE was defined as a composite measure of peripheral artery disease, non-fatal myocardial infarction, non-fatal stroke of any classification, and CV death (sudden cardiac death or death due to acute myocardial infarction, heart failure, stroke, CV procedures, CV hemorrhage, or other CV causes).

Stratification variables included age, sex, race, ethnicity, smoking status, and diabetes, as these are risk factors for both AD and CV events [20,21]. Information on age, sex, race, and ethnicity were obtained from the KPNC electronic health records (EHRs) [22]. Smoking status was obtained from the social history data in the EHR based on patient-reported behavior/lifestyle. Data sources used to identify diabetes included primary hospital inpatient discharge or outpatient-visit diagnoses of diabetes, any prescription of a diabetes-related medication, pharmacy use of insulin or oral hypoglycemic agents, laboratory values of hemoglobin $A_{1C}$ and glucose, and self-reporting on member health surveys [23]. Diabetes was identified from the KPNC diabetes registry, which gathers data from various components of the KPNC EHR and related clinical pharmacy and laboratory databases over time [24,25]. The KPNC Diabetes Registry uses a validated identification algorithm that is 99% sensitive [23].

### Statistical analysis

As the current study was descriptive, formal power and sample size calculations were not applicable. The number of patients with an incident event was divided by the total person-years of observation within the eligible AD cohort to estimate the IR of each CV event. Person-time was calculated by summing the number of years between the index date and the censor date. End dates were determined separately for each safety event and were the earliest of (1) the first instance of a safety event after the index date, (2) the first instance of disenrollment or death, or (3) the end of the observation period (December 31, 2018).

Estimated rates per 1000 person-years of follow-up and 95% confidence intervals (CIs) were calculated, and no data were imputed. IRs per 1000 person-years were reported by sex, race, ethnicity, smoking status, and diabetes risk factors for the entire cohort and by severity (moderate or severe). IRs per 1000 person-years were also reported by age group (12–17, 18–39, 40–64, or ≥65 years) for the combined moderate-to-severe AD group. Statistical analyses were performed using SAS version 9.4 (Cary, NC).

### Results

Of 34,405 health plan members in the KPNC database with an AD diagnosis between 2007 and 2018, 18,997 were aged ≥12 years. Using predefined selection criteria, 9608 health plan members with moderate-to-severe AD were identified. Medical records of 100 cases (50 moderate and 50 severe cases) were reviewed for adjudication. Of the 50 moderate cases adjudicated, 80% were definite cases, 18% were possible cases, and 2% were not a case. Of the 50 severe cases, medical chart adjudication determined that 86% were definite cases, 4% were possible cases, and 10% were not cases. Both the moderate and severe cases met the predetermined threshold of ≥80% agreement between cases and were deemed acceptable for disease severity classification. After additional exclusion criteria were applied, the final cohort comprised 8197 patients aged ≥12 years with moderate-to-severe AD (Fig 1). The mean age of the cohort was 39.4 years, and most patients (87.3%) had moderate AD (Table 1). Most patients were women (63.8%); 35.6% were Asian, and 8.1% were Black; 33.9% were non-Hispanic White, and 14.5% were Hispanic. More than half of patients were never smokers (57.8%), and 40.3% had a body mass index of <25 kg/m$^2$.

The IRs of CV events per 1000 person-years and number of events for each IR in the moderate-to-severe AD cohort are shown in Fig 2. The highest (with overlapping 95% CIs) IR was

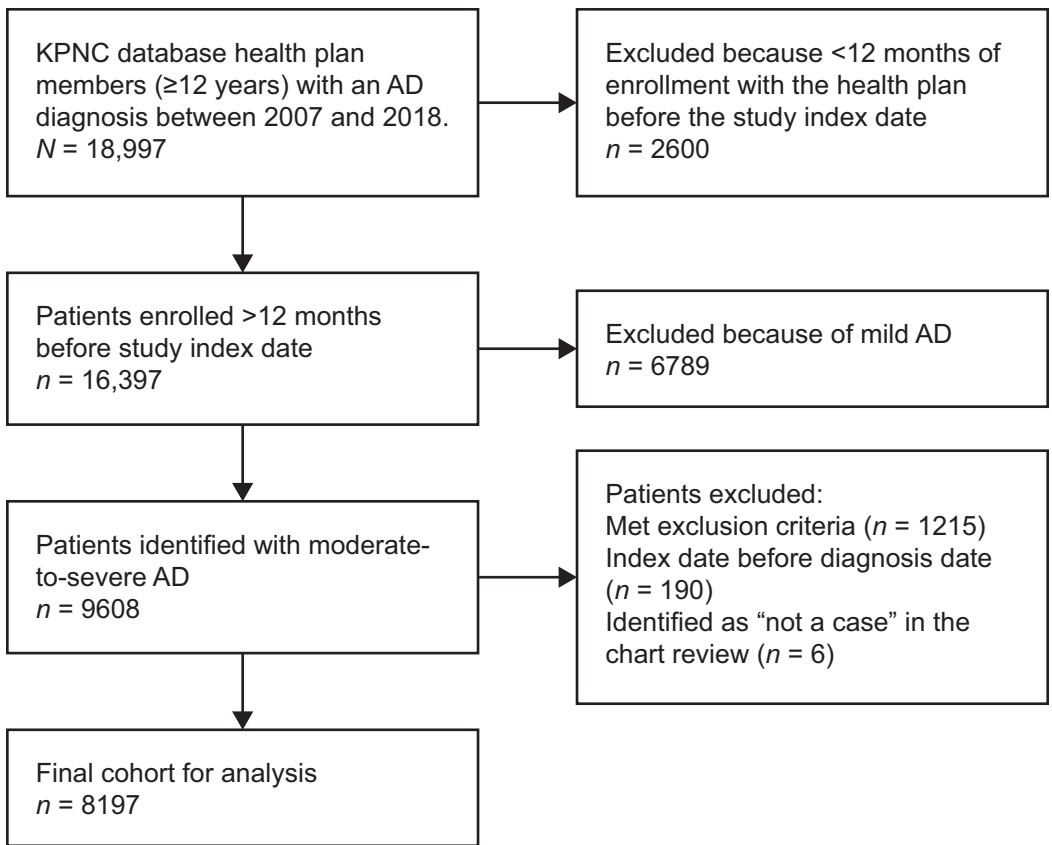

**Fig 1. Cohort selection.** AD, atopic dermatitis; KPNC, Kaiser Permanente Northern California.

for MACE, with 2.6 (95% CI, 2.1–3.2) events per 1000 person-years, followed by the IR for VTE of 2.0 (95% CI, 1.5–2.5) events per 1000 person-years; the incidence of DVT (1.6 [95% CI, 1.2–2.1] events per 1000 person-years) was higher than that of PE (0.7 [95% CI, 0.5–1.0] events per 1000 person-years).

IRs of CV events per 1000 person-years in the moderate-to-severe AD cohort generally were higher with older age, and more CV events occurred in older patients (≥65 years) compared with younger patients (12–17 years), as shown in Fig 2. MACE IRs were highest (with non-overlapping 95% CIs) in patients aged ≥65 years (14.1 [95% CI, 10.6–18.6]), followed by patients aged 40–64 years (2.9 [95% CI, 2.1–4.0]) and patients aged 18–39 years (0.3 [95% CI, 0.1–0.9]). No MACE events occurred in the 12- to 17-year age group. Similarly, IRs for VTE, DVT, and PE were higher in older compared with younger patients.

IRs for MACE per 1000 person-years were higher (with non-overlapping 95% CIs) among men than women (4.5 [95% CI, 3.4–5.8] vs. 1.6 [95% CI, 1.1–2.2]), former smokers than never smokers (4.3 [95% CI, 2.7–6.9] vs. 1.8 [95% CI, 1.3–2.5]), and those with diabetes than those without diabetes (9.3 [95% CI, 6.0–14.6] vs. 2.2 [95% CI, 1.7–2.8], respectively) (Table 2). VTE and DVT IRs were also higher (with non-overlapping 95% CIs) among those with diabetes than those without diabetes (5.1 [95% CI, 2.8–9.1] vs. 1.8 [95% CI, 1.4–2.3] and 4.1 [95% CI, 2.2–7.9] vs. 1.4 [95% CI, 1.1 vs. 1.9]), respectively. Although 95% CIs overlapped for these measures, IRs per 1000 person-years were numerically higher in men versus women for VTE (2.2 vs. 1.8) and DVT (1.7 vs. 1.5) and in former smokers versus never smokers for VTE (3.4 vs. 2.0) and DVT (2.7 vs. 1.7). In contrast, the incidence of PEs was higher with overlapping 95%

**Table 1. Demographic and baseline characteristics of the cohort of KPNC health plan members (2007–2018) aged ≥12 years with moderate-to-severe atopic dermatitis.**

| Characteristic | | Patients (*N* = 8197) |
|---|---|---|
| Age, mean (±SD), y | | 39.4 (±18.0) |
| Age category, *n* (%) | 12–17 years | 1147 (14.0) |
| | 18–39 years | 3183 (38.8) |
| | 40–64 years | 3058 (37.3) |
| | ≥65 years | 809 (9.9) |
| Sex, *n* (%) | | |
| | Women | 5229 (63.8) |
| | Men | 2966 (36.2) |
| | Data missing | 2 (<0.1) |
| Race, *n* (%) | | |
| | White | 2779 (33.9) |
| | Black | 664 (8.1) |
| | Asian | 2916 (35.6) |
| | Native American | 32 (0.4) |
| | Native Hawaiian or other Pacific Islander | 91 (1.1) |
| | Multiracial | 468 (5.7) |
| | Data missing/other | 1247 (15.2) |
| Ethnicity, *n* (%) | | |
| | Hispanic | 1192 (14.5) |
| | Non-Hispanic | 3669 (44.8) |
| | Data missing | 3336 (40.7) |
| AD severity, *n* (%) | | |
| | Moderate | 7158 (87.3) |
| | Severe | 1039 (12.7) |
| BMI, *n* (%) | | |
| | <25 kg/m$^2$ | 3303 (40.3) |
| | 25 to <30 kg/m$^2$ | 2180 (26.6) |
| | ≥30 kg/m$^2$ | 1677 (20.5) |
| | Data missing | 1037 (12.7) |
| Smoking, *n* (%) | | |
| | Yes/current | 462 (5.6) |
| | Quit/former | 1085 (13.2) |
| | Passive/secondhand | 92 (1.1) |
| | Never | 4737 (57.8) |
| | Data missing | 1821 (22.2) |
| Diabetes, *n* (%) | | |
| | Yes | 443 (5.4) |
| | No | 7754 (94.6) |

AD, atopic dermatitis; BMI, body mass index; KPNC, Kaiser Permanente Northern California; SD, standard deviation.

CIs for women than men (0.8 vs. 0.5) but otherwise followed the same trend with greater incidence in former smokers than never smokers (1.5 vs. 0.6) and in patients with diabetes versus without (1.8 vs. 0.6). No differences between patients with different AD severity were observed across IRs for any of the CV events examined (S3 Table).

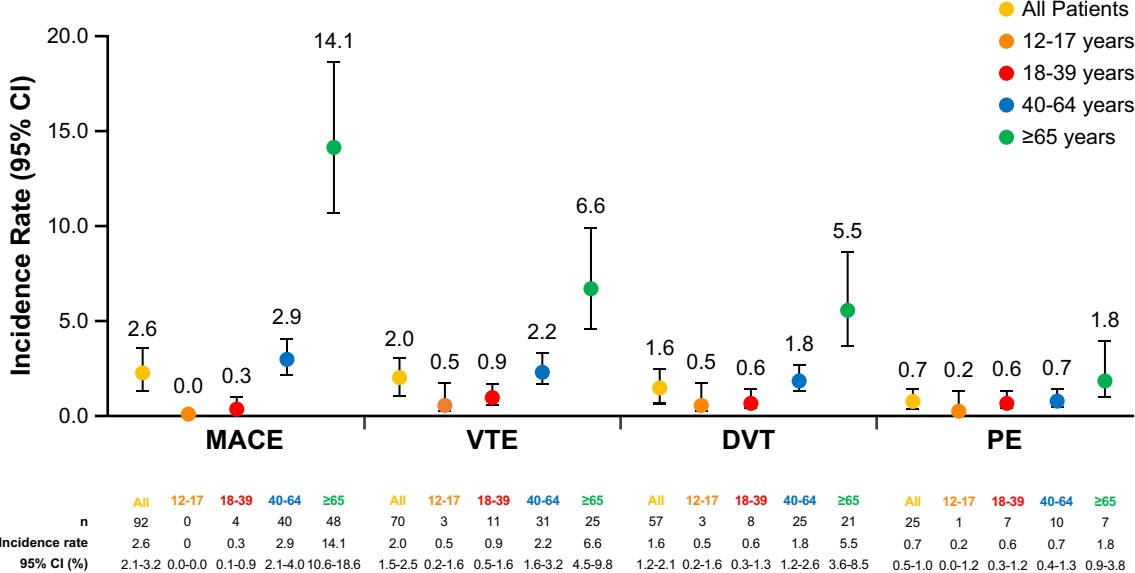

| | **All** | **12-17** | **18-39** | **40-64** | **≥65** | | **All** | **12-17** | **18-39** | **40-64** | **≥65** | | **All** | **12-17** | **18-39** | **40-64** | **≥65** | | **All** | **12-17** | **18-39** | **40-64** | **≥65** |
|---|---|---|---|---|---|---|---|---|---|---|---|---|---|---|---|---|---|---|---|---|---|---|---|
| n | 92 | 0 | 4 | 40 | 48 | | 70 | 3 | 11 | 31 | 25 | | 57 | 3 | 8 | 25 | 21 | | 25 | 1 | 7 | 10 | 7 |
| Incidence rate | 2.6 | 0 | 0.3 | 2.9 | 14.1 | | 2.0 | 0.5 | 0.9 | 2.2 | 6.6 | | 1.6 | 0.5 | 0.6 | 1.8 | 5.5 | | 0.7 | 0.2 | 0.6 | 0.7 | 1.8 |
| 95% CI (%) | 2.1-3.2 | 0.0-0.0 | 0.1-0.9 | 2.1-4.0 | 10.6-18.6 | | 1.5-2.5 | 0.2-1.6 | 0.5-1.6 | 1.6-3.2 | 4.5-9.8 | | 1.2-2.1 | 0.2-1.6 | 0.3-1.3 | 1.2-2.6 | 3.6-8.5 | | 0.5-1.0 | 0.0-1.2 | 0.3-1.2 | 0.4-1.3 | 0.9-3.8 |

**Fig 2. Incidence rates of cardiovascular events per 1000 person-years by age group in patients with moderate-to-severe atopic dermatitis.** CI, confidence interval; DVT, deep vein thrombosis; MACE, major adverse cardiovascular events; PE, pulmonary embolism; VTE, venous thrombotic event.

## Discussion

This retrospective study evaluated IRs of CV events in patients with moderate-to-severe AD. As composite measures, IRs of MACE and VTE were greatest per 1000 person-years in this patient population, followed by DVT and PEs. IRs of MACE were higher with increasing age groups, in men versus women, in former smokers versus patients who never smoked, and in patients with diabetes versus patients without diabetes; these observations are consistent with those for CV disease in the non-AD population [26–28]. Similarly, rates of VTE and DVT were higher among those with diabetes versus those without, which is consistent with observations in the non-AD population [29,30].

The IR for MACE observed in this study was 2.6 (95% CI, 2.1–3.2), and a difference was observed between women (1.6) and men (4.5), consistent with a higher risk of cardiac disease in men in the general population [31]. VTE and DVT IRs were markedly higher in this study than have been observed in the general US adult population (VTE: 2.0 [current study] vs. 1.1 [32]; DVT: 1.6 [current study] vs. 0.66 (32) per 1000 person-years. The IRs for PE (0.7 [current study] vs. 0.41 [32] per 1000 person-years) were similar [32].

Previous epidemiologic studies have suggested an association between AD and CV events. A recent meta-analysis found an increased relative risk of myocardial infarction, stroke, angina, and heart failure with increasing AD severity in cohort studies [9]. Another study found that inflammatory skin disease, such as AD, was associated with significantly higher odds of multiple CV disease risk factors, including hypertension and diabetes, compared to those without an inflammatory skin disease [21]. Biologically, AD may contribute to an increased risk of CV events due to the complex relationship between the immunologic pathways and immune-mediated inflammation in AD [33]. Studies have shown a strong association between systemic inflammatory diseases and CV disease; increased CV risk may be attributable to chronic inflammation [21,34], and patients with AD may have increased systemic immune activation as a key basis of comorbidity [8]. Furthermore, systemic treatments for AD may affect CV risk [9,10]. Previous work demonstrated that risk for CV events was

**Table 2. Incidence rates per 1000 person-years of cardiovascular events in the cohort of KPNC health plan members (2007–2018) aged ≥12 years with moderate-to-severe AD by sex, smoking status, and diabetes status.**

| Cardiovascular event | | Incidence rates per 1000 PY (95% CI) among patients with moderate-to-severe AD (N = 8197) |
|---|---|---|
| **MACE** | | 2.6 (2.1–3.2) |
| Sex | Women[a] | **1.6 (1.1–2.2)** |
| | Men | **4.5 (3.4–5.8)** |
| Smoking status | Smoker | 5.2 (2.7–9.9) |
| | Never smoker | 1.8 (1.3–2.5) |
| | Former smoker | 4.3 (2.7–6.9) |
| Diabetes status | Diabetes | **9.3 (6.0–14.6)** |
| | No diabetes | **2.2 (1.7–2.8)** |
| **Venous thrombotic event** | | 2.0 (1.5–2.5) |
| Sex | Women | 1.8 (1.3–2.4) |
| | Men | 2.2 (1.6–3.2) |
| Smoking status | Smoker | 1.1 (0.3–4.4) |
| | Never smoker | 2.0 (1.4–2.8) |
| | Former smoker | 3.4 (2.0–5.8) |
| Diabetes status | Diabetes | **5.1 (2.8–9.1)** |
| | No diabetes | **1.8 (1.4–2.3)** |
| **Deep vein thrombosis** | | 1.6 (1.2–2.1) |
| Sex | Women | 1.5 (1.1–2.1) |
| | Men | 1.7 (1.1–2.6) |
| Smoking status | Smoker | 1.1 (0.3–4.4) |
| | Never smoker | 1.7 (1.2–2.4) |
| | Former smoker | 2.7 (1.5–4.9) |
| Diabetes status | Diabetes | **4.1 (2.2–7.9)** |
| | No diabetes | **1.4 (1.1–1.9)** |
| **Pulmonary embolism** | | 0.7 (0.5–1.0) |
| Sex | Women | 0.8 (0.5–1.2) |
| | Men | 0.5 (0.3–1.1) |
| Smoking status | Smoker | 0.5 (0.1–3.9) |
| | Never smoker | 0.6 (0.3–1.0) |
| | Former smoker | 1.5 (0.7–3.3) |
| Diabetes status | Diabetes | 1.8 (0.7–4.9) |
| | No diabetes | 0.6 (0.4–1.0) |

AD, atopic dermatitis; CI, confidence interval; KPNC, Kaiser Permanente Northern California; MACE, major adverse cardiovascular event; PY, person-years.

[a]Items with non-overlapping confidence intervals are shown in bold.

greatest in patients receiving systemic treatment [10], although the authors did not distinguish between mild and moderate cases.

There are several limitations to the study that should be considered. The KPNC databases were missing information for some risk stratification variables, including approximately 22%, 12.7%, and 15.2% of data on smoking status, body mass index, and race, respectively; the absence of race data was largely owing to patients who reported Hispanic as their ethnicity and did not report race. In addition, these EHR data only included patients within the KPNC health care system, which may have limited generalizability to patients without access to health care or from other geographic regions. However, the KPNC dataset is representative of those with access to health care, as it provides insurance coverage and care to approximately 30% of all Northern Californians [18]. The generalizability of the findings may also be limited by the inclusion and exclusion criteria applied, as the cohort was originally designed to assess

background rates of CV events in patients with moderate-to-severe AD, mirroring patient populations included in recent AD clinical trial programs [35]. Additionally, the KPNC databases do not have detailed information on disease severity based on clinical examination using validated scoring lists and tools. Instead, moderate-to-severe AD disease was defined according to treatment, and the validity of our severity classification algorithm was adjudicated on a subset of patients by a board-certified dermatologist. Finally, this study did not assess data from patients with mild AD or include a healthy (non-AD) population for comparison.

## Conclusions

Among patients with moderate-to-severe AD in an integrated health care system, IRs per 1000 person-years were highest for MACE (2.6), followed by VTE (2.0), DVT (1.6), and PE (0.7). IRs were higher for older versus younger patients, patients with diabetes versus patients without, former smokers versus patients who had never smoked, and men versus women, except for PE, which was higher in women. Although patients with AD have been found to be at an increased risk for several CV safety events compared with patients without AD, prior studies did not report IRs of the events among patients with moderate-to-severe AD. Therefore, the current data provided by this retrospective cohort study of patients enrolled in the KPNC health system fill a gap in the scientific literature about the incidence of CV events in patients with moderate-to-severe AD and provide valuable information for ongoing clinical trials as well as clinicians caring for patients with moderate-to-severe AD.

## Supporting information

**S1 Table. Patient exclusion criteria.** AD, atopic dermatitis; ICD, International Classification of Diseases; JAK, Janus kinase; KP, Kaiser Permanente; KPNC, Kaiser Permanente Northern California; TB, tuberculosis.
(PDF)

**S2 Table. List of variables and operational definitions.** ICD, International Classification of Diseases; KPNC, Kaiser Permanente Northern California; MACE, major adverse cardiovascular event; MedDRA, Medical Dictionary for Regulatory Activities. [a]Asterisks used in ICD-9 and ICD-10 codes denote that all additional codes with numbers following the number to the left of the asterisk were also included.
(PDF)

**S3 Table. Incidence rates per 1000 Person-years of cardiovascular events in the cohort of KPNC health plan members (2007–2018) aged ≥12 years with moderate and severe AD by sex, smoking status, and diabetes status.** AD, atopic dermatitis; CI, confidence interval; MACE, major adverse cardiovascular event. [a]Items with non-overlapping confidence intervals for comparisons within the moderate AD population and within the severe AD population are shown in bold for comparisons.
(PDF)

## Acknowledgments

Editorial/medical writing support under the guidance of authors was provided by Kristine De La Torre, PhD, Marianna Johnson, PhD, and Natalie Barnes, BASLP, at ApotheCom, San Francisco, CA in accordance with Good Publication Practice (GPP3) guidelines (*Ann Intern Med.* 2015;163:461–464).

## Author Contributions

**Conceptualization:** Monique M. Hedderson, Charles P. Quesenberry, Jamie Geier.

**Data curation:** Fei Xu, Charles P. Quesenberry, Sneha Sridhar.

**Formal analysis:** Monique M. Hedderson, Fei Xu, Charles P. Quesenberry, Sneha Sridhar, Adina R. Lemeshow.

**Writing – original draft:** Monique M. Hedderson, Maryam M. Asgari, Fei Xu, Charles P. Quesenberry, Sneha Sridhar, Jamie Geier, Adina R. Lemeshow.

**Writing – review & editing:** Monique M. Hedderson, Maryam M. Asgari, Fei Xu, Charles P. Quesenberry, Sneha Sridhar, Jamie Geier, Adina R. Lemeshow.

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
