## [Decision Letter · Decision Letter 0]

11 Jul 2022

PONE-D-22-17307Rates of cardiovascular events among patients with moderate-to-severe atopic dermatitis in an integrated health care system: a retrospective cohort studyPLOS ONE

Dear Dr. Lemeshow,

Thank you for submitting your manuscript to PLOS ONE. After careful consideration, we feel that it has merit but does not fully meet PLOS ONE’s publication criteria as it currently stands. Therefore, we invite you to submit a revised version of the manuscript that addresses the points raised during the review process.

We look forward to receiving your revised manuscript.

Kind regards,

Dong Keon Yon, MD, FACAAI

Academic Editor

PLOS ONE

Journal Requirements:

"I have read the journal's policy and the

authors of this manuscript have the following competing interests: Monique M. Hedderson, Fei Xu, Charles P. Quesenberry, and Sneha Sridhar are full-time employees of Kaiser Permanente Northern California and received payment from Pfizer Inc. for the study discussed in this publication. Jamie Geier is a former employee of and shareholder of Pfizer Inc. Adina R. Lemeshow is an employee and shareholder of Pfizer Inc. Maryam M. Asgari received royalty payments from UptoDate for publications related to skin cancer and received grant funding to her institution from Pfizer Inc. for the study described in this publication. "

We note that you received funding from a commercial source: Pfizer Inc.

Additional Editor Comments:

I read it with great interest. Please address excellent comments of the reviewers.

#1. Please describe KPNC IRB approval number.

#2. Please cite the reference of AD definition. Previous paper (Prof Abuabara [UCSF]) will help a lot.

Silverwood RJ, Mansfield KE, Mulick A, Wong AYS, Schmidt SAJ, Roberts A, Smeeth L, Abuabara K, Langan SM. Atopic eczema in adulthood and mortality: UK population-based cohort study, 1998-2016. J Allergy Clin Immunol. 2021 May;147(5):1753-1763. doi: 10.1016/j.jaci.2020.12.001. Epub 2021 Jan 27. PMID: 33516523; PMCID: PMC8098860.

#3. In statistical section, please cite the statistical guideline (i.e., https://doi.org/10.54724/lc.2022.e3)

This is an excellent paper.

Reviewers' comments:

Reviewer's Responses to Questions

**Comments to the Author**

1. Is the manuscript technically sound, and do the data support the conclusions?

Reviewer #1: Yes

Reviewer #2: Partly

Reviewer #3: Yes

2. Has the statistical analysis been performed appropriately and rigorously? 

Reviewer #1: Yes

Reviewer #2: No

Reviewer #3: Yes

3. Have the authors made all data underlying the findings in their manuscript fully available?

Reviewer #1: Yes

Reviewer #2: No

Reviewer #3: Yes

4. Is the manuscript presented in an intelligible fashion and written in standard English?

Reviewer #1: Yes

Reviewer #2: Yes

Reviewer #3: Yes

5. Review Comments to the Author

Reviewer #1: Different scoring systems have been developed to determine the

Severity of atopic dermatitis, instead of judging the severity by treatment modality.

These validated scoring systems, is suited for clinical trials to give an accurate measurement and correlations with AD severity.

Reviewer #2: Rates of cardiovascular events among patients with moderate-to-severe atopic dermatitis in an integrated health care system: a retrospective cohort study

Why the authors and Pfizer chose this site for their studies. Please state all the reasons that apply to these answers.

Why no controls from the same center were not chose who have similar demographics and epidemiologic features such race, age, weight, high, smoking habits, other concomitant diseases, and or morbidities such alterations in triglycerides, lipids, high blood pressure, kind of diet etc. What is the statistical reason for not doing so? Kindly state all of them.

Why did the physical activity not include in the epidemiological retrospective study?

Why were other symptoms and signs of cardiovascular disease not choosing to be study in this study population? Such angina (chest pain) shortness of breath, irregular heartbeat, extreme fatigue, changes in their extremities, such as pain, swelling, tingling, numbness, coldness, arrythmias, some genetic cardiovascular anomalies?

The increase of cardiovascular events described by the authors show a tendency of increase in people that are older than 20 years old when in general the atopic dermatitis tends to be less severe. This seems to be a little less severe. This data has more a trend of cardiovascular events in the general populations. Please elaborate.

The racial increase in white and Asian seem to be a little high knowing that other underrepresented groups may go this Kaiser facility. How did the authors explain this?

This reviewer respectfully suggests that some epidemiologist at Kaiser permanent center where this study was performed to review properly the study, make suggestions for the improvement of epidemiological data.

Pfizer and the investigators with conflicts of interest also need to declare if this study is and or was going to be used for any medication that they are working to submit to the FDA a new drug application (NDA and what medication is in case of a positive answer?

Reviewer #3: Detail the title of the table 2. The titles of tables must be as complete as possible, including study location and period.

In table 1, place each characteristic (Age, sex, race, Ethnicity, AD severity, BMI, Smoking, Diabetes) in a first column. In a second column specify the variables for each characteristic.

Do the same in table 2: place each Cardiovascular event (MACE, Venous thrombotic event, Deep vein thrombosis, Pulmonary embolism) in a first column. In a second column, specify the variables women, man; smoker, never smoker, former smoker; diabetes, not diabetes.

As it is, everything in the same column, it is difficult to visualize.

As only AD cases used systemic medications, which could predispose to thrombotic events, I think the incidence of thrombotic events in moderate AD and severe AD should be analyzed separately. The authors of this research themselves make this comment between lines 262 and 264.

This study did not include a non-AD population for comparison. The authors could at least include in the introduction or discussion the incidence data reported in the literature for each thrombotic event in the healthy (non-AD) population.

6. PLOS authors have the option to publish the peer review history of their article (what does this mean?). If published, this will include your full peer review and any attached files.

Reviewer #1: No

Reviewer #2: No

Reviewer #3: **Yes: **Marilda Aparecida Milanez Morgado de Abreu

---

## [Author Response · Author response to Decision Letter 0]

12 Sep 2022

August 26, 2022

Point-by-Point Responses to the Editor and Reviewers

Editor’s Comments

Response: We have reviewed these templates carefully and made adjustments where necessary.

Response: We received a waiver of written informed consent from the IRB to use the patients’ electronic medical record data and thus, the patients did not provide informed consent for use of their data in this study. The data were fully anonymized before we accessed them. We have now added this information to the manuscript on page 4, lines 76–78: “A waiver of written informed consent to use patients’ electronic medical record data was provided by the IRB; thus, patients did not provide informed consent for use of their data in this study.” 

3. Thank you for stating the following in the Competing Interests section:"I have read the journal's policy and the authors of this manuscript have the following competing interests: Monique M. Hedderson, Fei Xu, Charles P. Quesenberry, and Sneha Sridhar are full-time employees of Kaiser Permanente Northern California and received grant funding from Pfizer Inc. for the study discussed in this publication. Jamie Geier is a former employee of and shareholder of Pfizer Inc. Adina R. Lemeshow is an employee and shareholder of Pfizer Inc. Maryam M. Asgari received royalty payments from UptoDate for publications related to skin cancer and received grant funding to her institution from Pfizer Inc. for the study described in this publication.”

We note that you received funding from a commercial source: Pfizer Inc. Please provide an amended Competing Interests Statement that explicitly states this commercial funder, along with any other relevant declarations relating to employment, consultancy, patents, products in development, marketed products, etc. 

Response: We have added the requested statement to our Competing Interests Statement. We have included all relevant declarations about employment, consultancy, patents, products in development, and marketed products.

Additional Comments from the Editor

I read it with great interest. Please address excellent comments of the reviewers.

#1. Please describe KPNC IRB approval number.

Response: The KPNC IRB approval number is IRBNet project no. 1340300.

#2. Please cite the reference of AD definition. Previous paper (Prof Abuabara [UCSF]) will help a lot. Silverwood RJ, Mansfield KE, Mulick A, Wong AYS, Schmidt SAJ, Roberts A, Smeeth L, Abuabara K, Langan SM. Atopic eczema in adulthood and mortality: UK population-based cohort study, 1998-2016. J Allergy Clin Immunol. 2021 May;147(5):1753-1763. doi: 10.1016/j.jaci.2020.12.001. Epub 2021 Jan 27. PMID: 33516523; PMCID: PMC8098860.

Response: We have added this reference in the Introduction (page 3 line 48; ref number 7).

#3. In statistical section, please cite the statistical guideline (i.e., https://doi.org/10.54724/lc.2022.e3)

Response: Thank you for sharing this article. We read it carefully and found that it does not seem to address the types of statistical analysis performed in our paper. We do not suggest including this in our manuscript and have not cited it in our revised draft.

This is an excellent paper.

Reviewer Comments

Comments to the Author

1. Is the manuscript technically sound, and do the data support the conclusions?

Reviewer #1: Yes

Reviewer #2: Partly

Reviewer #3: Yes

2. Has the statistical analysis been performed appropriately and rigorously? 

Reviewer #1: Yes

Reviewer #2: No

Reviewer #3: Yes

3. Have the authors made all data underlying the findings in their manuscript fully available?

Reviewer #1: Yes

Reviewer #2: No

Reviewer #3: Yes

Response: As noted in our Data Availability Statement (pages 8–9, lines 167–174), “The data used for this study contain protected health information and access is protected by the KPNC Institutional Review Board. Individual level data may not be made publicly available due to IRB and privacy concerns. Data are available from the Kaiser Permanente Division of Research for researchers who meet the criteria for access to confidential data. For more information about data access and criteria for access to confidential data, please contact Kaiser Permanente Division of Research: DOR.IRB.Submissions@kp.org. All other relevant data are within the paper.”

We believe that this statement addresses the PLOS data policy.

4. Is the manuscript presented in an intelligible fashion and written in standard English?

Reviewer #1: Yes

Reviewer #2: Yes

Reviewer #3: Yes

5. Review Comments to the Author

Reviewer #1: 

Different scoring systems have been developed to determine the Severity of atopic dermatitis, instead of judging the severity by treatment modality. These validated scoring systems, is suited for clinical trials to give an accurate measurement and correlations with AD severity.

Response: Thank you for this comment. The definition of severity is described in our paper as follows (page 5, lines 98–102): “Moderate AD cases were defined as necessitating prescribed topical therapy or phototherapy, and severe AD cases were defined as necessitating prescribed systemic treatment (i.e., oral corticosteroids, cyclosporine, azathioprine, methotrexate, mycophenolic acid/mycophenolic mofetil, interferon-gamma, alitretinoin, rituximab, ustekinumab, omalizumab, dupilumab, or apremilast).” Because this was a retrospective observational study, we relied on treatment modality to assess severity.

In addition, severity classification was confirmed by adjudication as stated in the paper: “A medical record review of 100 cases (50 moderate and 50 severe AD cases) was conducted by a board-certified dermatologist (author MMA) to adjudicate and confirm the classification of AD, including AD severity based on established diagnostic criteria” (page 6, lines 129–131) and “A priori, a threshold of ≥80% agreement between the cases identified by study parameters and medical record review was deemed acceptable for disease severity classification” (page 6, lines 127–129).

Reviewer #2: 

Rates of cardiovascular events among patients with moderate-to-severe atopic dermatitis in an integrated health care system: a retrospective cohort study

Why the authors and Pfizer chose this site for their studies. Please state all the reasons that apply to these answers.

Response: The rationale is stated in the Study design subsection of the Materials and methods section in the manuscript (page 4, lines 82–85): “KPNC is an integrated health care delivery system that provides comprehensive health care and pharmaceutical benefits to approximately 30% of a large and diverse community-based population residing in Northern California. The sociodemographic and health characteristics in the service area are very similar to the general population in Northern California [18].” We think that this cohort reasonably represents the population of patients with AD with access to health care.

Why no controls from the same center were not chose who have similar demographics and epidemiologic features such race, age, weight, high, smoking habits, other concomitant diseases, and or morbidities such alterations in triglycerides, lipids, high blood pressure, kind of diet etc. What is the statistical reason for not doing so? Kindly state all of them.

Response: The original purpose of this analysis was to provide context for the rates of safety events observed in the AD clinical program; the Kaiser rates are in a background population rather than in a population treated with a specific medicine. We compared the CV rates observed in the clinical trial to those provided in the current Kaiser analysis to determine whether the rates in the trial were those expected in the general AD population—or whether they were higher or lower. Although a comparison group would have helped us make inferences about populations with and without AD, that was not the original purpose of these analyses. These analyses were done to provide context for clinical trials.

Why did the physical activity not include in the epidemiological retrospective study?

Response: Data on physical activity were not available in the electronic health records.

Why were other symptoms and signs of cardiovascular disease not choosing to be study in this study population? Such angina (chest pain) shortness of breath, irregular heartbeat, extreme fatigue, changes in their extremities, such as pain, swelling, tingling, numbness, coldness, arrythmias, some genetic cardiovascular anomalies?

Response: Data on the symptoms noted above were not available in the electronic health records. As noted above, the original purpose of this analysis was to provide context for the rates of safety events observed in the AD clinical program. The cardiovascular outcomes evaluated in this study were selected because they were relevant to the clinical program.

The increase of cardiovascular events described by the authors show a tendency of increase in people that are older than 20 years old when in general the atopic dermatitis tends to be less severe. This seems to be a little less severe. This data has more a trend of cardiovascular events in the general populations. Please elaborate.

Response: Although AD may tend to be less severe with older age, the risk of cardiovascular events increases with age; this is reflected in the reviewer’s observation and is addressed in our paper on page 15, lines 241–244): “IRs of MACE were higher with increasing age groups, in men versus women, in former smokers versus patients who never smoked, and in patients with diabetes versus patients without diabetes; these observations are consistent with those for CV disease in the non-AD population.” This statement is supported by reference no. 26 (Virani et al. Circulation 2021;143:e254–743), and we have now included 2 additional references for further support of this statement (reference no. 27, Cushman et al. Am J Med. 2004;117:19–25; reference no. 28, Naess et al. J Thromb Haemost. 2007;5:692–699).

The racial increase in white and Asian seem to be a little high knowing that other underrepresented groups may go this Kaiser facility. How did the authors explain this?

Response: In our cohort of KPNC Health Plan members aged ≥12 years with moderate-to-severe AD, 35.6% were Asian by race and 33.9% were White. The relatively high proportion of Asian patients in this cohort is consistent with recent studies reporting that AD is more common in Asian patients than in many other racial/ethnic groups. In addition, northern California has a large population of Asian residents. As noted in the Discussion section (page 16, lines 269–271), “…the absence of race data was largely owing to patients who reported Hispanic as their ethnicity and did not report race”. This may partially explain the unexpected racial distribution as noted by the reviewer. 

This reviewer respectfully suggests that some epidemiologist at Kaiser permanent center where this study was performed to review properly the study, make suggestions for the improvement of epidemiological data.

Response: Dr. Monique M. Hedderson (first author) is a researcher at Kaiser Permanente and has an PhD in epidemiology. Co-author Dr. Charles Quesenberry holds a PhD in biostatistics and is the Associate Director of the Biostatistics section at Kaiser Permanente Division of Research. Co-authors Drs. Adina R. Lemeshow and Jamie Geier also hold PhDs in epidemiology. These authors ensured that the analyses were conducted properly and in a methodologically rigorous way.

Pfizer and the investigators with conflicts of interest also need to declare if this study is and or was going to be used for any medication that they are working to submit to the FDA a new drug application (NDA and what medication is in case of a positive answer)?

Response: This study is not being used to support approval of any new drugs.

Reviewer #3:

Detail the title of the table 2. The titles of tables must be as complete as possible, including study location and period.

Response: We have updated the title of Table 1 (page 10, lines 194–195) to “Demographic and Baseline Characteristics of the Cohort of KPNC Health Plan Members (2007–2018) Aged ≥12 Years With Moderate-to-Severe Atopic Dermatitis”. We have updated the title of Table 2 (page 13, lines 231–233) to “Incidence Rates per 1000 Person-Years of Cardiovascular Events in the Cohort of KPNC Health Plan Members (2007–2018) Aged ≥12 Years With Moderate-to-Severe AD by Sex, Smoking History, and Diabetes Status.”

In table 1, place each characteristic (Age, sex, race, Ethnicity, AD severity, BMI, Smoking, Diabetes) in a first column. In a second column specify the variables for each characteristic. Do the same in table 2: place each Cardiovascular event (MACE, Venous thrombotic event, Deep vein thrombosis, Pulmonary embolism) in a first column. In a second column, specify the variables women, man; smoker, never smoker, former smoker; diabetes, not diabetes. As it is, everything in the same column, it is difficult to visualize.

Response: We have made the requested changes to Tables 1 and 2 (pages 10–11, lines and pages 13–15).

As only AD cases used systemic medications, which could predispose to thrombotic events, I think the incidence of thrombotic events in moderate AD and severe AD should be analyzed separately. The authors of this research themselves make this comment between lines 262 and 264.

Response: We have added the requested data in a new S3 Table to the supplement. This table is cited in the aforementioned text.

This study did not include a non-AD population for comparison. The authors could at least include in the introduction or discussion the incidence data reported in the literature for each thrombotic event in the healthy (non-AD) population.

Response: The following is included in the Discussion of the current manuscript draft (pages 15–16, lines 247–252): “The IR for MACE observed in this study was 2.6 (95% CI, 2.1–3.2), and a difference was observed between women (1.6) and men (4.5), consistent with a higher risk of cardiac disease in men in the general population [31]. VTE and DVT IRs were markedly higher in this study than have been observed in the general US adult population (VTE: 2.0 [current study] vs. 1.1 [32]; DVT: 1.6 [current study] vs. 0.66 [32] per 1000 person-years. The IRs for PE (0.7 [current study] vs. 0.41 [32] per 1000 person-years) were similar [32].”

6. PLOS authors have the option to publish the peer review history of their article (what does this mean?). If published, this will include your full peer review and any attached files. Do you want your identity to be public for this peer review? For information about this choice, including consent withdrawal, please see our Privacy Policy.

Reviewer #1: No

Reviewer #2: No

Reviewer #3: Yes: Marilda Aparecida Milanez Morgado de Abreu

---

## [Decision Letter · Decision Letter 1]

19 Oct 2022

PONE-D-22-17307R1Rates of cardiovascular events among patients with moderate-to-severe atopic dermatitis in an integrated health care system: A retrospective cohort studyPLOS ONE

Dear Dr. Lemeshow,

Thank you for submitting your manuscript to PLOS ONE. After careful consideration, we feel that it has merit but does not fully meet PLOS ONE’s publication criteria as it currently stands. Therefore, we invite you to submit a revised version of the manuscript that addresses the points raised during the review process.

We look forward to receiving your revised manuscript.

Kind regards,

Dong Keon Yon, MD, FACAAI

Academic Editor

PLOS ONE

Additional Editor Comments:

Reviewers and I have serious concerns about the author's participation in Pfizer. Please describe it to reduce the ethical problems in Method section.

Reviewers' comments:

Reviewer's Responses to Questions

**Comments to the Author**

1. If the authors have adequately addressed your comments raised in a previous round of review and you feel that this manuscript is now acceptable for publication, you may indicate that here to bypass the “Comments to the Author” section, enter your conflict of interest statement in the “Confidential to Editor” section, and submit your "Accept" recommendation.

Reviewer #3: All comments have been addressed

2. Is the manuscript technically sound, and do the data support the conclusions?

Reviewer #3: Yes

3. Has the statistical analysis been performed appropriately and rigorously? 

Reviewer #3: Yes

4. Have the authors made all data underlying the findings in their manuscript fully available?

Reviewer #3: Yes

5. Is the manuscript presented in an intelligible fashion and written in standard English?

Reviewer #3: Yes

6. Review Comments to the Author

Reviewer #3: The suggested changes were made by the authors. Thus, I think the manuscript is suitable for publication.

7. PLOS authors have the option to publish the peer review history of their article (what does this mean?). If published, this will include your full peer review and any attached files.

Reviewer #3: **Yes: **Marilda Aparecida Milanez Morgado de Abreu

---

## [Author Response · Author response to Decision Letter 1]

26 Oct 2022

October 26, 2022

Dong Keon Yon, MD, FACAAI

Academic Editor

PLOS ONE

Dear Dr. Yon,

Thank you for your continued feedback on our manuscript, “Rates of cardiovascular events among patients with moderate-to-severe atopic dermatitis in an integrated health care system: A retrospective cohort study” (PONE-D-22-17307), and for continuing to liaise with the PLOS ONE reviewers. We welcome the feedback and greatly appreciate comments from Marilda Aparecida Milanez Morgado de Abreu indicating that all suggested changes have been made and that they believe the manuscript is suitable for publication.

Regarding the feedback indicating concerns about the authors’ participation with Pfizer, we can confirm that the ‘Competing Interests Statement’ we submitted electronically along with the manuscript explicitly identifies Pfizer as a commercial funder and includes all relevant declarations related to employment, consultancy, patents, products in development, marketed products. This statement also confirms that Pfizer funding does not alter authors’ adherence to all PLOS ONE policies on sharing data and materials by including the following statement. For further clarity, the following has been written into the Materials and Methods section (Page 4 Lines 78-84): 

While Pfizer is a commercial funder of this study, at no point did this interfere with the full and objective presentation, peer review, editorial decision-making, or publication of research data submitted to PLOS in this manuscript. Pfizer products are not evaluated in this study, and no medical value claims are made regarding Pfizer products. Pfizer and all authors declare the results of this study will not be used for any medication that Pfizer is working to submit to the regulatory authorities as part of a new drug application, including the US Food and Drug Administration.

For full transparency, we also included fully updated conflict of interest information for all authors in the original and resubmitted version. If there are any other changes necessary for compliance reasons, please do not hesitate to let us know and we will remedy immediately.

We sincerely thank the editor and reviewers for their continued efforts, time and attention to our manuscript. These comments have helped us strengthen our manuscript. 

We are hopeful this updated manuscript is suitable for publication in PLOS ONE and look forward to hearing from you at your convenience.

Sincerely,

Adina Lemeshow

Adina.lemeshow@pfizer.com

---

## [Editor Report · Decision Letter 2]

28 Oct 2022

Rates of cardiovascular events among patients with moderate-to-severe atopic dermatitis in an integrated health care system: A retrospective cohort study

PONE-D-22-17307R2

Dear Dr. Lemeshow,

We’re pleased to inform you that your manuscript has been judged scientifically suitable for publication and will be formally accepted for publication once it meets all outstanding technical requirements.

Kind regards,

Dong Keon Yon, MD, FACAAI

Academic Editor

PLOS ONE
---

## [Editor Report · Acceptance letter]

7 Nov 2022

PONE-D-22-17307R2 

Rates of cardiovascular events among patients with moderate-to-severe atopic dermatitis in an integrated health care system: A retrospective cohort study 

Dear Dr. Lemeshow:

I'm pleased to inform you that your manuscript has been deemed suitable for publication in PLOS ONE. Congratulations! Your manuscript is now with our production department. 

Kind regards, 

on behalf of

Dr. Dong Keon Yon 

Academic Editor

PLOS ONE